# Accelerated Life Testing of Biodegradable Starch Films with Nanoclay Using the Elongation Level as a Stressor

**DOI:** 10.3390/foods13203333

**Published:** 2024-10-21

**Authors:** Theofilos Frangopoulos, Apostolos Ketesidis, Anna Marinopoulou, Athanasios Goulas, Dimitrios Petridis, Vassilis Karageorgiou

**Affiliations:** Department of Food Science and Technology, International Hellenic University, P.O. Box 141, 57400 Thessaloniki, Greece; thfrangopoulos@gmail.com (T.F.); apostolosketesidis@gmail.com (A.K.); amarinop@ihu.gr (A.M.); agoulas@ihu.gr (A.G.); petridis@ihu.gr (D.P.)

**Keywords:** accelerated life testing, biodegradable starch films, simple linear model, acceleration factor, failure prediction

## Abstract

An attempt was made to evaluate the elongation level as a stressor on biodegradable starch films reinforced with nanoclay using a simple linear model. A total of 120 film units were subjected to increasing elongation levels and the exact break time of the failed units was monitored. Nine different attempts were made to fit the data distribution and the lognormal distribution was chosen as the most suitable because it resulted in the lowest values of the regression fit indices −2LL, AICc and BIC. Following the selection of the best fit, it was, generally, observed that an increase in the elongation level resulted in the decreasing exact break time of the films. Among several models, the best fit was provided by the simple linear model. Based on this model, the acceleration factor was estimated, and it was shown that it increased exponentially while increasing the elongation level. Finally, the probability of failure and the hazard rate of the film units as a function of the elongation level were estimated, demonstrating the applicability of this method as a tool for food packaging film failure prediction.

## 1. Introduction

Nowadays, economic globalization is present in all human activities, and its presence in the food chain cannot be an exception [1]. This means that food distribution ceases to have distance barriers and products produced on one side of the world can reach consumers on the other side [2]. But although this reality offers new commercial and economic perspectives, it also puts pressure on science and the food industry to adapt to the requirements that a product may have to travel under adverse conditions for a long period of time until it reaches the consumer [3,4]. Following this logic, an attempt was made to find those factors that could compromise the quality of a food product under prolonged shelf life or adverse conditions until it reaches the market [5]. Some of the main factors that degrade products are the storage temperature [6], which can increase enzymatic [7] and microbiological [8] deterioration, illumination [6], mechanical damage due to hitting or excessive shear stresses [9] and processing temperature [10]. These factors can reduce the lifespan of the products and must be taken into account in the design.

The most basic factor that protects a food product under adverse conditions and must keep it intact until it reaches the consumer is its packaging [11]. Food product packaging is not just a means of transport and marketing but must also protect the food product from moisture absorption, photo-oxidation, mechanical damage and oxygen uptake [12]. Because moisture is an important factor in food spoilage and both high and low moisture food products are susceptible to deterioration, either because of drip loss or moisture absorption, the use of moisture absorbers or modified atmosphere packaging (MAP) could provide a solution [13]. Photo-oxidation can lead to off-flavors and off-odors as much as deterioration of natural pigments; therefore, food packaging systems could benefit from the incorporation of UV blocking agents [14]. Since oxygen is a factor promoting the deterioration of both the nutritional and sensory quality of foods, packaging systems could implement MAP or oxygen scavenging technologies [15]. Finally, active packaging strategies have also been tested for their efficiency against deteriorating factors in relation to foods, resulting in their shelf life extending. More specifically, active packaging strategies involve compounds with antioxidant and antimicrobial activity, which interact with the food and the external environment, halting the deteriorating effects of microbial spoilage and oxidation, thus extending the food product’s shelf life [16]. All these developments in the food packaging sector are expected to have a tremendous impact on food waste reduction, since, if packaging innovations are implemented more extensively in the food industry, it is expected that, by 2050, 100 million tons of food will be saved (amounting to 50% of food waste at the retail and consumer level) due to the extension of food shelf life [17]. However, since packaging itself is subject to degradation due to mechanical damage or temperature, the limits of durability must be determined because in many cases these limits determine the shelf life of products. If the shift of the food packaging industry from conventional plastic to biodegradable packaging made of biopolymer components is also taken into account, an extra pressure is put on the food packaging industry to address all the above-mentioned food downgrades during production and distribution.

The continuing restrictions on the use of conventional plastics as food packaging and the increasing awareness of consumers about reducing their carbon footprint are stimulating the need to find alternative polymers of natural origin that could act as packaging [18]. Probably, one of the most important legal developments in this area was the Chinese waste import ban of 2017, which drove countries from all over the world to find alternatives for their plastic waste, including biodegradable packaging strategies, which resulted to decreasing plastic waste [19]. Also, another important legal development in this direction was the prohibition or at least restriction of single-use plastics by a directive of the European Commission in 2018 [20]. Undoubtedly, in this direction, one of the most efficient biopolymers is starch, which possesses the advantages of its abundance in nature, the feasibility of being formed into films, as well as the possibility of manufacturing food packaging with a variety of physicochemical and sensory properties [21]. However, due to the natural origin of starch, biodegradable starch packages are characterized by poor mechanical properties as well as high water vapor permeability compared to their conventional polymer counterparts. A solution to this issue is to reinforce the starch matrix with nanoclays such as montmorillonite. During intercalation, the polymer chains penetrate in between the nanoclay platelets and nanoclays act as junction zones, strengthening the starch matrix structure [21]. For example, it was observed that the increase in the tensile strength of polylactic acid films reinforced with nanoclay was due to the hydrogen bonds between the polylactic acid and the nanoclay [22]. A similar explanation (new hydrogen bonds between starch hydroxyl, carboxymethyl cellulose carboxyl and hydroxyl and montmorillonite hydroxyl groups) was given for the enhancement of the tensile strength in starch–carboxymethyl cellulose–montmorillonite films [23].

It is, therefore, crucial to find a tool to predict the failure rate of food packaging, let alone biodegradable starch films, under the action of stress factors over long periods of time in order to allow the industry to modify the shelf life of the food product appropriately. In general, normal use of products will result in a few units failing or degrading appreciably [24] and, as is well understood for other materials [25], attempts to analyze the lifespan and failure rate of food packaging subjected to normal stress conditions could be very difficult, if not impossible, and would require a long period of time as well as many raw materials. This is why the use of accelerated life testing (ALT), which involves subjecting certain units of starch films to specific increasing levels of stress, acting as a stressor, has been proposed. Originally, in the book by Nelson [26], the use of temperature as a stressor in accelerated life testing of insulation tape using the Arrhenius model was described [26]. Most often in the literature, accelerated life testing is performed under a single stressing agent, at constant or variable values [27], although combinations of stressing agents, usually involving temperature as one of them, have been attempted [24]. Examples of statistical models for acceleration include the general time-transformation functions, the scale-accelerated failure-time models, the proportional hazard regression model and the nonconstant *σ* regression model [24]. Moreover, specific models have been proposed in the case of certain stressors, e.g., the Arrhenius model or extensions of it, such as the Eyring model [25] for temperature or the inverse power relationship for voltage [24]. Recently, Nelson (2024) reviewed the constant and varying stress tests, the analyses of the model fits to the data and the prediction of life distribution [28]. Statistical practices have been proposed as an alternative to engineering intuition regarding the development of the design methodology of accelerated life testing [29].

In the thin film food packaging sector, accelerated life testing has been applied to solid whole egg powder [30], young radish kimchi [31], dried pumpkin snacks [32] and coffee [33] or liquid olive oil [34] and orange juice [35]. However, in all these and similar studies comparing thin films as food packaging materials, the effect of temperature as a stressor on the food and not the packaging material is investigated. There is only one case where accelerated life testing has been applied to food packaging materials and this is the application of the linear model to hydrolyzed corn starch films using temperature and humidity as stressors [36]. Nevertheless, the two stressors were not studied as a combined effect and the model used only the temperature effect.

The aim of this study was to evaluate the use of the elongation level as a stressor on the lifespan of starch films reinforced with nanoclay through the use of a linear regression model and to predict both the failure rate and the time required for the failure of a certain percentage of experimental units at different levels of the stressor. The elongation was chosen as a stressor because all packaging materials undergo mechanical loads that could potentially compromise their integrity, posing a threat to the safety of the food product that is contained within. In a previous study [37], the forward selection of the most important variables was used in multiple regression analysis to construct a destructive degradation model describing the fatigue behavior of biodegradable starch-based food packaging films reinforced with nanoclay using the elongation level as a stressor. The experimental data from that study defined the strength limits of the specimens and the experimental elongation levels in such a way that there were films that survived but also films that broke, at all levels, which are key prerequisites for calculating the acceleration factor. Although the acceleration factor has been calculated for various materials in several studies through multiple stressors, this is the first time that the acceleration factor was calculated through the use of mechanical stress and, more specifically, elongation.

In 2015, all the United Nations Member States adopted the 2030 Agenda for Sustainable Development, which set 17 Sustainable Development Goals (SDGs) aiming to achieve peace and prosperity for people and the planet, now and into the future. Our research can contribute to achieving the following goals:▪#8 Decent work and economic growth and #12 Responsible consumption and production: The production of starch films could contribute to the circular economy and sustainability, particularly if the starch source is food (e.g., legumes and rice) that is rejected as unfit for human and animal consumption from the food industry.▪#9 Industry, innovation and infrastructure: The proposed methodology does not require expensive reagents or equipment; therefore, it could upgrade the technological capabilities of industrial sectors even in developing countries.▪#14 Life below water: The development of fully biodegradable packaging will greatly contribute to reducing the marine life-threatening plastic and microplastic ocean pollution.

## 2. Materials and Methods

### 2.1. Materials

Two types of legumes (lentils (*Lens culinaris*) and chickpea (*Cicer aretinum*)) were used to isolate starch. The process was carried out at a pilot plant scale and is described in the study by Frangopoulos et al. (2023) [38]. The moisture and protein content of the starches was lower than 12 and 1.5%, respectively [38]. X-ray diffraction shows that legume starches are characterized by a C-type pattern [39]. The molecular weights of legume amylose range from 1.65 × 10^5^ to 3.12 × 10^5^ [39], and the average molecular weights of amylopectin from legume starches are greater than 1.9 × 10^7^ [40,41]. Marinopoulou et al. (2016) [42] reported that the viscosity average molecular weight of pea amylose is 2.7 × 10^5^, as determined using an Ubbelohde capillary viscometer [42]. Glycerol of 99% purity was bought from Carlo Erba Reagents (Cornaredo, Italy). Montmorillonite (15A Montmorillonite Bis(hydrogenated tallow alkyl) dimethyl, salt with bentonite, <80 nm APS, 99% purity, 1.98 g/cm^3^ density, <3% moisture was bought from Nanoshel LG (Sundran, Punjab, India) and was used as the nanoclay.

### 2.2. Film Casting

The casting method was used to produce the nanoclay-reinforced biodegradable starch films, as described in a previous study [37]. Water, starch and glycerol were weighted to prepare an aqueous dispersion containing 7% wt. starch and 35% wt. glycerol based on dry starch. The dispersion was heated inside a water bath at 80 °C for 30 min to gelatinize the starch. Continuous mixing was supplied via a waterproof magnetic stirring plate. After starch gelatinization and the preparation of the thermoplastic starch dispersion, montmorillonite was added (1% wt. based on dry starch). To achieve proper hydration, the montmorillonite dispersion had been stirred overnight and sonicated for 15 min at 15,000 kHz. Following the montmorillonite addition, the temperature was raised to 90 °C and the dispersion was kept at this temperature for 15 min. This step ensured the intercalation of the starch chains between the montmorillonite sheets. Casting on 18 cm × 11 cm Plexiglass trays was performed immediately after heating, making sure that the dispersion was spread evenly on the trays. Finally, the trays were transferred to an oven operating at 45–50 °C and the dispersion was dried under an air current for ~24 h. This time period allowed the easy peeling of the films from the trays when drying was completed. Before any further testing, the films’ moisture was equilibrated by storing them for 10 days in the lab at ambient temperature.

### 2.3. Thickness Measurement

A digital electronic caliper (TESA, Brown & Sharpe Instruments, North Kingstown, RI, USA) was used to measure the thickness of the films. A total of 12 measurements (4 equally distant, vertical points and 3 equally distant, horizontal measuring points at each vertical point) were performed for each film.

### 2.4. Tensile Test

A tensile test was conducted following the Standard Test Method for Tensile Properties of Thin Plastic Sheeting (ASTM D882-10) [43]. The film specimens had dimensions of 100 mm × 15 mm (length × width) and were cut with a lancet from the equilibrated films. The test was performed at ambient temperature (23 °C) using a texture analyzer (TA-XT Plus, Stable Microsystems Ltd., Godalming, UK) and the separation speed of the probe was 50 mm min^−1^.

### 2.5. Experimental Design

After the preliminary experiments, the film sample that was selected for analysis contained 7% wt. starch, 35% wt. glycerol and 1% wt. nanoclay based on a dry starch basis. Five increasing elongation levels (5, 12, 16, 18 and 23%) were chosen and 24 film units were assigned to each. The preliminary experiments determined the maximum elongation level of 23%, because nearly all the films broke beyond it. The elongation level of 5% was set as a reference level, since all or almost all the units passed through this stage intact. The intermediate levels were selected, since they were the levels where peaks in the frequency distribution of a previous destructive degradation model were observed [37]. All the units were subjected to the aforementioned tensile test to determine the exact break time. The exact break times of the film units at each elongation level are shown in Table 1. The goal of this study was to establish the best-fitting linear life distribution model in order to estimate the acceleration factor (if the data contributed to its suitability) and the predictive ability of the model at all levels, even at low levels of film elongation, outside the limits of the accelerated stress, and also at high exact break times. All the statistical analyses were performed using the JMP 17.2 statistical software.

## 3. Results and Discussion

Other properties of the films (i.e., tensile properties, water vapor permeability, opacity index and L*a*b* color values, antimicrobial activity against mesophilic and psychrotrophic bacteria and biodegradation under simulated composting conditions) have been described in a previous publication [38]. The average thickness of the 120 films that were subjected to the tensile test in the present study was 0.21 mm (values ranged between 0.19 mm and 0.23 mm).

According to the arrangement of the data in Table 1, it can be observed that, in general, the film’s exact break time decreased when the elongation level increased, and this decrease is described on the scatterplot of the break time as a function of the elongation level (Figure 1). The lines of the 10%, 50% and 90% break time percentiles make the downward trend more apparent, particularly with the help of the density distribution of the data per action level. In the graph, only the 5% and 23% elongation levels differentiate obviously from each other judging from the points distributions. This finding is supported by the statistical significance of the change in the exact break time between the elongation levels according to the non-parametric Kruskal–Wallis test of homogeneity between levels (*p* < 0.0001) (Table 2). This test indicates that at least one elongation level differs significantly from the others in terms of the median break time. The decreasing trend in the survival time as the stressor levels increased was also shown in the work of Escobar and Meeker [24], where voltage stress was applied to mylar-polyurethane insulation using the inverse power-lognormal model. In the same study, the decreasing trend was also apparent when different relative humidity levels were used as a stressor on printed circuit boards’ failure [24].

The lognormal distribution was chosen as the most suitable after nine consecutive attempts to fit the data distribution, since it yielded the lowest values of the regression fit indices -2LL, AICc and BIC (Table 3) [44].

The Akaike and Bayesian information criteria (AIC and BIC) are alternative expressions of the multiple regression specification R^2^ with increasing popularity in data with non-normal distributions and in logistic regressions, and they are used frequently in cases where the selection of variables leads to the synthesis and comparison of complex models and with different numbers of inclusion terms. These indices have a similar explanatory significance to that of the R^2^ coefficient and express a better fit to lower values. The model with the smallest AIC value is comparatively considered to provide the best fit. In models with a small sample size (≤10–15), the corrected AICc is used. Note that the AICc and BIC indices are chosen to compare different models and smaller values indicate a better fit of the data. The indices simply assess the quality of each model relative to the others, thus facilitating the best choice.

Using the lognormal distribution to the data, a regression analysis was applied by examining the location (mean values of the levels) and scale (regression slopes at the elongation levels) parameters for their equality. The regression model, i.e., the simple linear model, that accepts a single common slope for all the elongation levels and equality of the means between levels was selected for further processing. This model provides a low AICc index (1047.58) (Table 4), although it is slightly higher than the model with a different location, but it is the only one showing statistical significance compared to the other two (*p* < 0.0001) (Table 5).

The simple linear model is characterized by three parameters: the slope b_1_ of the fitting line, which expresses the acceleration rate of the elongation stressor, the value of the slope b_0_ on the *Y*-axis (exact break time) and the standard deviation σ. The coefficient b_1_ has a negative value and expresses the decreasing trend of the ln exact break time by 0.225 (this means that the exact break time decreases by 1252 times) when the elongation increases by 1% at a time (Table 6). A similar calculation of these parameters, but in this case for a second-degree polynomial model (therefore β_0_, β_1_ and β_2_), is described for a cell counting reagent subjected to accelerated temperature conditions in order to predict its degradation [45].

Therefore, the equation of the simple linear model that describes the correlation between the ln exact break time (*μ*) and the elongation level is as follows:(1)μ=9.084−0.225×elongation level

The validity of the simple linear model is stressed by the strong linear trend of the elements in the Cox–Snell residual probability graph (Figure 2) and by the uniform dispersion of the points in the standardized residuals graph with the predicted values of the response Y (exact break time) (Figure 3).

The simple linear model exclusively allows the estimation of the acceleration factor, which takes a specific value for each level of elongation compared to the reference elongation level (5%). The acceleration factor (*AF*) of each elongation level is derived by dividing the predicted quantile value (*p_b_*) of the exact break time at the 5% reference level by the predicted value (*p_s_*) at the elongation level (*i*) using the antilogarithms. The same method for obtaining the acceleration factor is also found in the literature, but mainly assuming the Arrhenius model for temperature as a stressor [6,25].
(2)$$AF=\frac{p_{b}}{p_{s}}=\frac{e^{\mu_{5\%}}}{e^{\mu_{i}}}$$

In other applications, similar equations have been proposed for the calculation of the AF. For example, when the temperature [46], humidity [47] or irradiation [48] was used as a stressor to estimate the degradation of food products, the AF was defined as the ratio of the reaction velocities of the accelerated over market test conditions. When the temperature was used as a stressor to estimate the migration of 2-aminobenzamide from polyethylene terephthalate beverage bottles, the AF was defined as the ratio of the time frame to reach a certain concentration at normal over accelerated conditions [49].

Applying Equation (2) to all four levels of increasing action of the elongation results in the table of the acceleration factor per elongation level (Table 7), in which the increase in the acceleration factor even at a minimal incremental change, as is the case from level 16% to level 18%, where the acceleration factor increased from 11.25 to 17.46, is easily seen. Consequently, the acceleration factor profiler (Figure 4), showing the exponential increase in the acceleration factor with an increasing elongation level, is generated. This increase in the acceleration factor while increasing the stress levels was also observed when different levels of voltage stress were applied to the insulation for generator armature bars [24]. In that study, when the voltage stress was 130 V/mm, the acceleration factor was around 2, but when the voltage stress increased to 155 V/mm, the acceleration factor increased to around 10. In the present study, the shift from 12 to 23% elongation stress caused approximately a 12-fold increase in the acceleration factor (Table 7).

Figure 5 describes the cumulative probability of failure of the films F(t) (distribution profiler). The example of the exact break time of 3000 s was chosen as the required time interval to predict failure by which, according to the data, almost all of the films have already broken. Adjusting the elongation level percentage at the examined investigated levels from Figure 5, the table of failure probabilities for the exact break time of 3000 s at the selected levels is obtained (Table 8). The high percentage of failed films at an elongation level of 12% (86.5%), which intensifies significantly at an elongation level of 16% (95.7%) and almost completely at a level of 23% (99.7%), is clearly depicted. As can be seen from Table 8, even at the highest investigated stress duration (3000 s), under a constant stress, the percentage of failed film units is relatively low at elongations between 5% and 12% (51.3% and 86.5%, respectively). A similar increase in the cumulative probability along with the increase in the stressor was also observed when different stress factors were applied to train traffic [50]. When the temperature was used as a stressor in starch films’ accelerated shelf-life testing, a failure probability of 99% was reached at 9 months of testing time at 20 °C [36]. Obviously, a direct comparison cannot be made between the present study and this one, since different stressors were used.

In the quantile profiler of the elongation stress (Figure 6), the decreasing trend of the exact break time can easily be seen, as the elongation level increases and approaches 154 s for 50% probability of failure at the elongation level of 18%. Setting the probability of failure at 50%, the table of the elongation action period at all levels of interest is obtained (Table 9), where the extreme difference in time between the 5% reference level (2864.8 s) and the 23% level (50.1 s) is remarkable. Thus, the quantile profiler answers the question of the median survival time at which 50% of the tested units will pass intact at each elongation level, always assuming a lognormal distribution of data.

In the hazard profiler (Figure 7), aka the non-parametric failure rate [5], a linear change in the instantaneous failure rate is observed while increasing the elongation action. By setting the exact break time to the near maximum level of film destruction that occurs at 3000 s, the hazard rate per level of elongation is calculated. At the maximum elongation level (23%), an instantaneous hazard rate of 0.07% per second of action of the stressor was observed. Compared to the reference elongation level of 5%, this maximum elongation level instantaneous hazard rate was about 3.5 times higher (Table 10). The application of the linear model in accelerated life testing using the temperature and relative humidity as stressors allowed the prediction of the percentage of non-degraded hydrolyzed corn starch films under normal temperature conditions [36].

Based on the accelerated testing theory, a unit operating under increasing levels of stress will experience the same failure conditions as when operating under normal stress conditions, i.e., the difference lies in the faster occurrence of the failure event. In other words, any underlying chemical or physical process that leads to product failure is expected to occur exactly similarly at lower stress levels and the only difference is the change in the time interval (time scale) between the occurrence of failure. Therefore, if it is known how the units perform under stress, as well as the appropriate transfer of the time scale to the lower stress levels, the distribution of the unit failure time at the lower levels quantitatively and the validity statistically can be calculated. This time transformation constitutes a simple acceleration model that for practical and understandable purposes is expressed in terms of constant multiples of the time scale known as accelerators. Specifically, for the failure time at one stress level, the extrapolation of the same time to the next stress level can be calculated, thus implying a linear acceleration, which for two failure times is expressed by t_u_ = AF t_s_, where t_u_ is the failure time under normal stress conditions and t_s_ under higher stress conditions. In summary, the data analysis concerning the experimental elongation of packaging films showed that the simple regression equation with a common slope at all the elongation levels and the parameter b_0_ (slope height) can be fitted to the linear model. This model allowed:▪the successful calculation of the AF using four experimental levels of percentage elongation, always compared to the normal use level of 5%,▪the estimation of the failure rate of the films per elongation level at the maximum values of the breakage time, which is particularly high at the 16–23% (>95%) levels,▪the estimation of the elongation action time per elongation level with a minimum at 50.1 s at 23% elongation and a maximum at the reference level of 5% (2864.8 s),▪the estimation of the risk rate, which is four times the risk at 23% elongation than at the 5% level.

## 4. Conclusions

The goal of the presented research was to evaluate the use of the elongation level as a stressor on the lifespan of starch films reinforced with nanoclay through the use of the simple linear model and to predict both the failure rate at different levels of the stressor and the time required for the failure of a certain percentage of experimental units at different levels of the stressor. In general, it was observed that an increase in the elongation level caused a decrease in the film’s exact break time. The simple linear regression that is described by a single common slope for all the elongation levels and the equality of the means between levels was selected for further processing. Based on this model, the failure of the films was predicted. The findings of this study show that it is efficient to use the elongation level as a stressor of biodegradable films’ failure rate by applying a simple linear model. Also, they enable the opportunity to predict useful parameters, such as the percentage of failed films at different exact break times and elongation levels. However, it should be considered that the study was performed using a certain recipe for the starch, glycerol and montmorillonite content. The variability of the concentrations of the ingredients as well as the different types of nanoclays that could be used, all having an impact on the film properties, constitute a challenge in terms of establishing an experimental design to include all the aforementioned combinations. In addition, a realistic shelf-life prediction is limited by the use of only one stressor, since there are several other factors (e.g., temperature, humidity, UV irradiation) that contribute to the degradation of starch films. In future experiments, the elongation level will be combined with other stressors using the Eyring relationship to give a more realistic prediction of the endurance of biodegradable food packaging films based on starch. A potential experimental design could include several levels for combinations of stressors, such as the temperature (in a temperature regulated oven), humidity (using different saturated salt solutions) or elongation (as presented in this study), applied to starch films with different species (montmorillonite, bentonite, halloysite, kaolinite) and concentrations of nanoclay.

## Figures and Tables

**Figure 1 foods-13-03333-f001:**
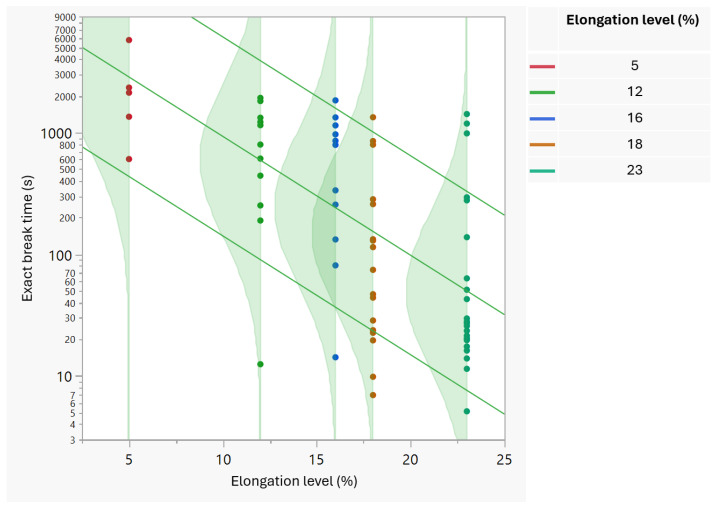
Scatterplot of the exact break time as a function of the elongation level.

**Figure 2 foods-13-03333-f002:**
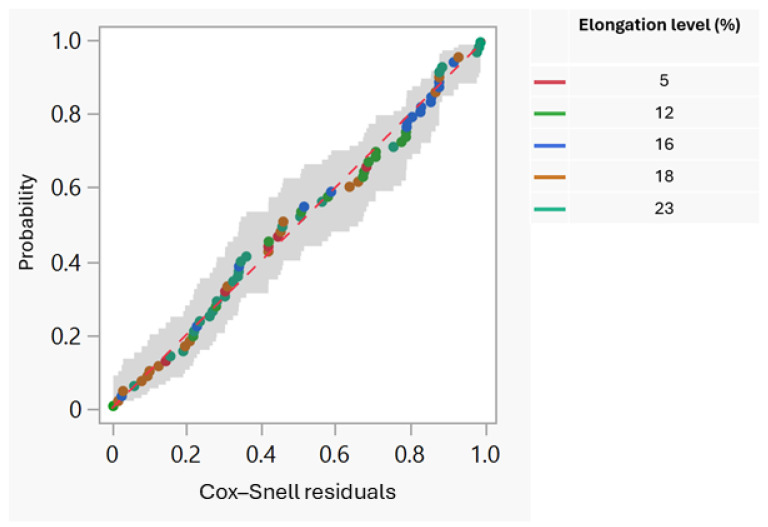
Cox–Snell residual probability graph.

**Figure 3 foods-13-03333-f003:**
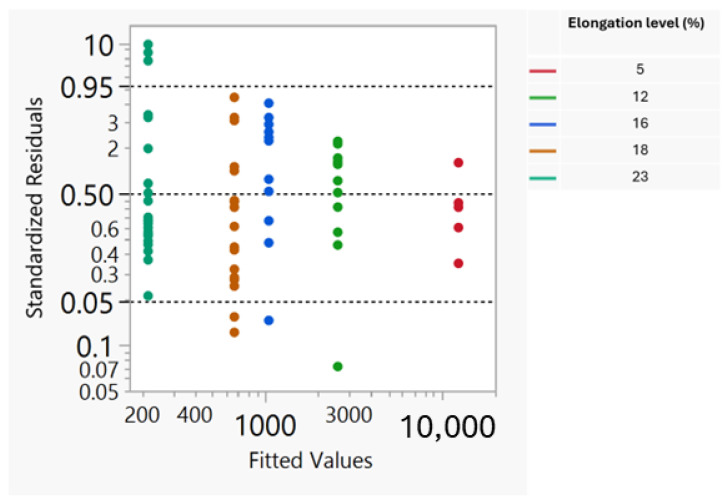
Standardized residuals versus the predicted values of the exact break time.

**Figure 4 foods-13-03333-f004:**
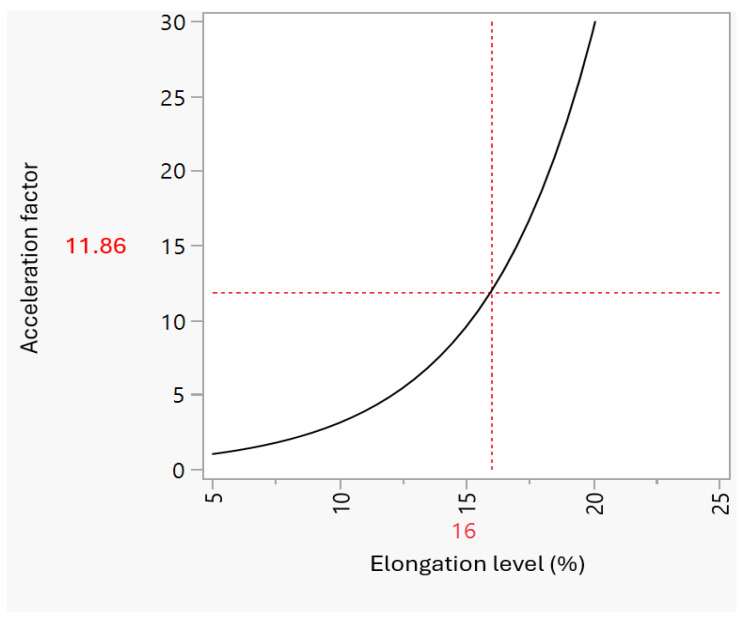
Acceleration factor profiler as a function of the elongation level.

**Figure 5 foods-13-03333-f005:**
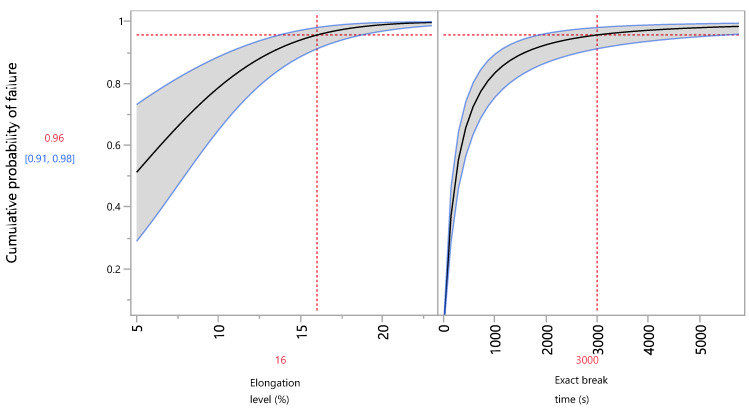
Distribution profiler of the cumulative probability of failure vs. the elongation level and exact break time.

**Figure 6 foods-13-03333-f006:**
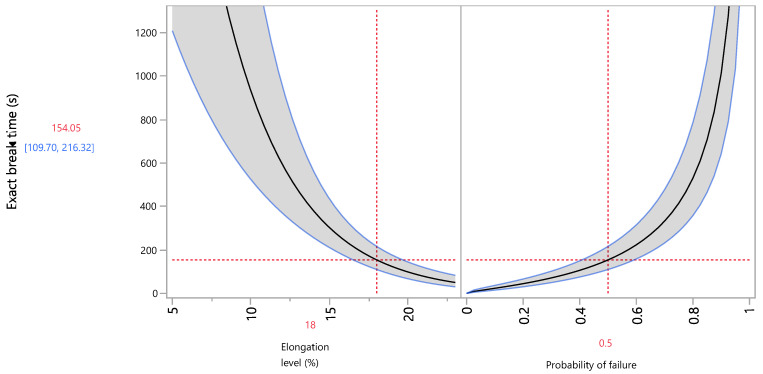
Quantile profiler of the exact break time vs. the elongation level and probability of failure.

**Figure 7 foods-13-03333-f007:**
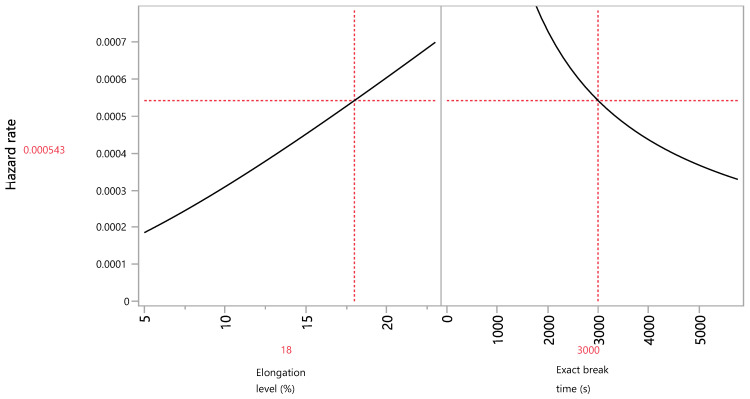
Hazard profiler as a function of the elongation level and exact break time.

**Table 1 foods-13-03333-t001:** Exact break times for each film unit based on the elongation level.

Elongation Level (%)	Number of Broken Films	Mean Exact Break Time (s)	Standard Deviation	Min	Max
5	5	2439.4	1982.4	606.0	5766.2
12	14	1012.3	644.7	12.5	1931.5
16	15	801.3	538.2	14.2	1840.7
18	17	245.9	381.3	7.0	1337.1
23	23	206.4	405.4	5.1	1421.0

**Table 2 foods-13-03333-t002:** Non-parametric Kruskal–Wallis test of homogeneity.

χ^2^	DF	*p*
31.75	4	<0.001

**Table 3 foods-13-03333-t003:** Reliability indices of the data fit.

Distribution	-2LL	AICc	BIC
Lognormal	1041.24	1047.58	1054.14
Loglogistic	1042.92	1049.26	1055.83
Weibull	1046.73	1053.06	1059.63
Fréchet	1054.36	1060.70	1067.26
Exponential	1058.99	1063.16	1067.60
LEV	1141.69	1148.03	1154.60
Logistic	1157.01	1163.35	1169.92
Normal	1177.46	1183.79	1190.36
SEV	1226.51	1232.85	1239.42

**Table 4 foods-13-03333-t004:** Reliability indices of the models’ fit.

Model	-2LL	AICc	BIC	Number of Parameters
No effect	1076.50	1080.60	1085.10	2
Regression	1041.24	1047.58	1054.14	3
Separate location	1033.73	1046.98	1059.55	6
Separate location and scale	1029.95	1053.47	1073.02	10

**Table 5 foods-13-03333-t005:** Statistical information on the models’ significance.

Description	L-R Chi-Square	DF	*p*
No effect vs. Regression	35.26	1	<0.0001
Regression vs. Separate location	7.51	3	0.0574
Separate location vs. Separate location and scale	3.78	4	0.4413

**Table 6 foods-13-03333-t006:** Simple linear model parameter estimates.

Parameter	Estimate	Std Error	Lower 95%	Upper 95%
β_0_	9.084	0.598	7.912	10.257
β_1_	−0.225	0.033	−0.290	−0.159
σ	1.469	0.121	1.233	1.706

**Table 7 foods-13-03333-t007:** Acceleration factor per elongation level.

Elongation Level (%)	Μ	Acceleration Factor
5	7.98	1.00
12	6.44	4.66
16	5.56	11.25
18	5.12	17.46
23	4.02	52.46

**Table 8 foods-13-03333-t008:** Probability of failure at each elongation level for a 3000 s exact break time, including the 95% confidence intervals (CIs).

Elongation Level (%)	Probability	Lower CI	Upper CI
5	0.513	0.289	0.732
12	0.865	0.768	0.929
16	0.957	0.912	0.981
18	0.979	0.948	0.992
23	0.997	0.987	0.999

**Table 9 foods-13-03333-t009:** Exact break time at each elongation level for the 0.5 probability of failure, including the 95% confidence intervals (CIs).

Elongation Level (%)	Exact Break Time (s)	Lower CI	Upper CI
5	2864.8	1208.3	6792.0
12	593.7	369.3	954.4
16	241.5	171.4	340.3
18	154.1	109.7	216.3
23	50.1	30.1	83.3

**Table 10 foods-13-03333-t010:** Instantaneous hazard rate at each elongation level for a 3000 s exact break time.

Elongation Level	Hazard Rate (%)
5	0.02
12	0.04
16	0.05
18	0.05
23	0.07

## Data Availability

The original contributions presented in the study are included in the article, further inquiries can be directed to the corresponding author.

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
