# Peer review of "Accelerated Life Testing of Biodegradable Starch Films with Nanoclay Using the Elongation Level as a Stressor"

_foods, 2024, doi:10.3390/foods13203333_

Round 1
Reviewer 1 Report
Comments and Suggestions for Authors
Below you can find my notes on this article:
- The discussion on the significance of food packaging as a protective barrier (Lines 42-44) is underdeveloped. It would benefit from including more recent statistics or data on the impact of packaging on food waste reduction.
- The transition to biodegradable films (Lines 54-58) lacks detail about the limitations of conventional plastics beyond environmental concerns, such as regulatory restrictions. Including this context would strengthen the argument for biodegradable alternatives.
- The rationale for selecting elongation as the stressor is not well-articulated.
- The description of film preparation could include more details about potential sources of variability in film thickness and how these were controlled. It is unclear whether steps were taken to ensure uniformity in film thickness across samples, which could affect the reliability of the results.
- The method for nanoclay incorporation (Lines 122-125) mentions stirring and sonication but does not specify the parameters used during sonication (e.g., power settings, duration), which are critical for reproducibility.
- The selection of elongation levels (5%, 12%, 16%, 18%, and 23%) appears arbitrary without clear justification. There should be an explanation of why these specific levels were chosen and how they relate to real-world mechanical stress conditions experienced by packaging films.
- While the manuscript discusses choosing the lognormal distribution for the model fit, it does not explain why other potential models, such as Weibull or exponential distributions, were less suitable. More detail on the criteria for model selection would provide a stronger justification for using the lognormal model.
- The statistical significance discussion (Lines 194-204) lacks detail about the implications of the different AICc values presented. It would be beneficial to elaborate on why the chosen model is preferred over others based on these values.
- The explanation of the acceleration factor calculation (Lines 234-245) should provide more context about how this factor compares to previous studies that used other stressors, like temperature or humidity.
- The discussion of failure probabilities across different elongation levels (Lines 263-265) should include a more detailed comparison to previous studies involving ALT of biodegradable films.
- The conclusions are somewhat generic and do not address the study's limitations or areas for future improvement. There should be a more explicit discussion of the limitations related to using a single stressor and potential variability in film preparation.
- Future work suggestions (Lines 315-316) should be more specific, outlining possible experimental designs that incorporate multiple stressors or different types of nanomaterials to validate the findings.
- The manuscript could benefit from additional citations related to the reinforcement mechanisms of nanoclay in biopolymer films.
below you can find my notes on this article:
- The discussion on the significance of food packaging as a protective barrier (Lines 42-44) is underdeveloped. It would benefit from including more recent statistics or data on the impact of packaging on food waste reduction.
- The transition to biodegradable films (Lines 54-58) lacks detail about the limitations of conventional plastics beyond environmental concerns, such as regulatory restrictions. Including this context would strengthen the argument for biodegradable alternatives.
- The rationale for selecting elongation as the stressor is not well-articulated.
- The description of film preparation could include more details about potential sources of variability in film thickness and how these were controlled. It is unclear whether steps were taken to ensure uniformity in film thickness across samples, which could affect the reliability of the results.
- The method for nanoclay incorporation (Lines 122-125) mentions stirring and sonication but does not specify the parameters used during sonication (e.g., power settings, duration), which are critical for reproducibility.
- The selection of elongation levels (5%, 12%, 16%, 18%, and 23%) appears arbitrary without clear justification. There should be an explanation of why these specific levels were chosen and how they relate to real-world mechanical stress conditions experienced by packaging films.
- While the manuscript discusses choosing the lognormal distribution for the model fit, it does not explain why other potential models, such as Weibull or exponential distributions, were less suitable. More detail on the criteria for model selection would provide a stronger justification for using the lognormal model.
- The statistical significance discussion (Lines 194-204) lacks detail about the implications of the different AICc values presented. It would be beneficial to elaborate on why the chosen model is preferred over others based on these values.
- The explanation of the acceleration factor calculation (Lines 234-245) should provide more context about how this factor compares to previous studies that used other stressors, like temperature or humidity.
- The discussion of failure probabilities across different elongation levels (Lines 263-265) should include a more detailed comparison to previous studies involving ALT of biodegradable films.
- The conclusions are somewhat generic and do not address the study's limitations or areas for future improvement. There should be a more explicit discussion of the limitations related to using a single stressor and potential variability in film preparation.
- Future work suggestions (Lines 315-316) should be more specific, outlining possible experimental designs that incorporate multiple stressors or different types of nanomaterials to validate the findings.
- The manuscript could benefit from additional citations related to the reinforcement mechanisms of nanoclay in biopolymer films.
Reviewer 2 Report
Comments and Suggestions for Authors
Some Insights
Abstract:
Introduction
- Please include in the introduction the Sustainable Development Goals (SDGs) that your research can contribute to. You can find the goals at https://sdgs.un.org/goals.
2. Materials and Methods
2.1 Please provide additional information about the starch, such as its molar weight.
2.2
- How can you control humidity?
- What is the effect of moisture equilibration during the 10-day storage period at room temperature? Since moisture can influence the film's properties, why wasn’t the film packaged immediately for storage instead of allowing it to equilibrate first?
2.3 Why did you take 12 measurements?
2.4 What about the repetitions? How can you ensure that the lancet used to cut the film does not influence the analysis, as it could potentially create small fractures that affect the results? Additionally, how did you control the temperature in your lab, since I know it can influence the analyses?
2.5 Why were one hundred twenty films used? I didn’t understand the rationale behind this number.
3. Line 165: Did you mean that the 120 films had a thickness between 0.19 and 0.23 mm?
Results and Discussion
I need the answers to these questions to better understand your work.
Reviewer 3 Report
Comments and Suggestions for Authors
This manuscript reports the accelerated life testing of biodegradable starch films with nanoclay using the elongation level as a stressor. The work is interesting, and it could be acceptted for publication after a major revision as follows:
1.Why choose this model? How to ensure the reliability of the model and the accuracy of predictions?
2.What are the reasons for selecting 5 levels of increasing elongation in the experimental design? Why not continue to increase the selection of values greater than 23%.
3.There are a few spelling errors and other errors like line 191 “[30](30)(30)[30](30) 30[30] 3030[30, 31]” in the manuscript.
4.Please unify the format of the references.
5.The author should supplement the manuscript with the predictive ability and accuracy level of models from other research teams, in order to better demonstrate the significance of this work.
Comments on the Quality of English LanguageThere are a few spelling errors and other errors like line 191 “[30](30)(30)[30](30) 30[30] 3030[30, 31]” in the manuscript.
Round 2
Reviewer 2 Report
Comments and Suggestions for Authors
Authors, thank you for the corrections. Everything is ok now.